# Brown Seaweed *Egregia menziesii*’s Cytotoxic Activity against Brain Cancer Cell Lines

**DOI:** 10.3390/molecules24020260

**Published:** 2019-01-11

**Authors:** Tatiana Olivares-Bañuelos, Anllely G. Gutiérrez-Rodríguez, Rodolfo Méndez-Bellido, Ricardo Tovar-Miranda, Omar Arroyo-Helguera, Claudia Juárez-Portilla, Thuluz Meza-Menchaca, Luis E. Aguilar-Rosas, Luisa C. R. Hernández-Kelly, Arturo Ortega, Rossana C. Zepeda

**Affiliations:** 1Instituto de Investigaciones Oceanológicas, Universidad Autónoma de Baja California, Ensenada, Baja California 22860, Mexico; tatiana.olivares@uabc.edu.mx (T.O.-B.); aguilarl@uabc.edu.mx (L.E.A.-R.); 2Programa de Doctorado en Ciencias Biomédicas, Universidad Veracruzana, Xalapa, Veracruz 91190, Mexico; anlly_grizett@hotmail.com; 3Instituto de Ciencias Básicas, Universidad Veracruzana, Av. Dr. Luis Castelazo Ayala s/n. Col., Industrial Ánimas, Xalapa, Veracruz 91190, Mexico; rmendez@uv.mx (R.M.-B.); rtovar@uv.mx (R.T.-M.); 4Instituto de Salud Pública, Universidad Veracruzana, Av. Dr. Luis Castelazo Ayala s/n, Col., Industrial Ánimas, Xalapa, Veracruz 91190, Mexico; oarroyo@uv.mx; 5Centro de Investigaciones Biomédicas, Universidad Veracruzana. Av. Dr. Luis Castelazo Ayala s/n. Col., Industrial Ánimas, Xalapa, Veracruz 91190, Mexico; cljuarez@uv.mx; 6Laboratorio de Genómica Humana, Facultad de Medicina, Universidad Veracruzana, Médicos y Odontólogos S/N, Col., Unidad del Bosque, Xalapa, Veracruz 91010, Mexico; tmeza@uv.mx; 7Laboratorio de Neurotoxicología, Departamento de Toxicología, Centro de Investigación y de Estudios Avanzados del Instituto Politécnico Nacional, Ciudad de México 07300, Mexico; lcr@cinvestav.mx (L.C.R.H.-K.); arortega@cinvestav.mx (A.O.)

**Keywords:** marine algae, cancer, seaweeds properties, anti-tumor activity, cell-toxicity

## Abstract

Brown seaweeds contain bioactive compounds that show anti-tumorigenic effects. These characteristics have been repeatedly observed in the *Lessoniaceae* family. *Egregia menziesii*, a member of this family, is distributed in the North Pacific and its properties have been barely studied. We evaluated herein the cytotoxic and anti-proliferative activity of extracts of this seaweed, through toxicity assay in *Artemia salina* and lymphocytes, and MTT proliferation assay, in Bergmann glia cells, 3T3-L1 and brain cancer cell lines. *E. menziesii*’s extracts inhibited the spread of all the tested cell lines. The hexane extract showed the highest cytotoxic activity, while the methanol extract was moderately cytotoxic. Interestingly, seaweed extracts displayed a selective inhibition pattern. These results suggest that *E. menziesii*’s extracts might be good candidates for cancer prevention and the development of novel chemotherapies due to its highest cytotoxicity in transformed cells compare to glia primary cultures.

## 1. Introduction

In recent years, a wide variety of biological activities have been characterized in marine products. Among the different marine sources for bioactive substances, seaweeds are of special importance due to the rich, varied, and underexploited amount of bioactive substances they contain. Seaweeds are used broadly as a source of food and either industrial or pharmaceutical products [1,2,3]. Properties such as fungicidal [4], antimicrobial [5], antimalarial [6], antimycobacterial [7], and antiviral [8] activities have been demonstrated to be present in algae. A number of diverse metabolites with important bioactivities have been isolated from these organisms with particular attention in phlorotannins, sulphated polysaccharides, polyphenols, carotenoids, peptides, and sterols or polyunsaturated fatty acids [9].

Particularly, brown seaweeds of the *Lessoniaceae* family are rich in polysaccharides including alginic acids, laminarans, fucoidans, phlorotannins, and diterpenes [10]. *Ecklonia cava*, a brown alga of this family that contains phlorotannins, has been reported to display diverse biological activities such as antioxidant effects [11], and its immunomodulatory properties have been associated with antidiabetic [12], antihypertensive [13], anti-inflammatory [14], radioprotective [15], and anti-proliferative [16] effects. *Egregia menziesii* is another marine brown alga belonging to *Lessoniaceae* family, distributed in the North Pacific from Alaska to Mexico, whose biological activity is being investigated due to its close phylogenetic relationship with *Ecklonia cava*.

In 2012, the World Health Organization (WHO) estimated more than 14 million new cancer cases, and at least 8.2 million deaths from this cause [17]. Cancers of the central nervous system (CNS) are among the 15 most common types of cancer in both men and women. Only in 2012, 256,000 new cases of the CNS cancers and 189,000 deaths, representing 1.8% of new cancers and 2.3% of cancer deaths were reported [18]. Specifically, glioblastomas are the most frequent and aggressive type of tumors, and comprises more than 70% of all brain cancers [19].

Currently, cancer treatment is based on chemotherapy and radiotherapy, which have demonstrated to cause a wide range of side effects. Even though chemotherapy is the most used, especially to treat patients with metastasis, this therapy shows low specificity to suppress cancer cells proliferation without a significant damage to normal cells [20]. In this context, compounds from natural sources with anti-proliferative activity represent an important and novel alternative to treat several types of cancer. Brown algae are gaining importance due to the proven anti-proliferative properties of sulfated polysaccharides and fucoidans, which are abundant in these organisms [9]. Phloroglucinol’s derivatives isolated from *Ecklonia cava* have a significant anti-proliferative activity over MCF7 human cancer cells [16]. Likewise, extracts from the brown algae *Laminaria japonica* inhibit proliferation of human hepatocellular carcinoma (BEL7402) and murine leukemic cells (P388) [21]. Moreover, some seaweeds compounds have been tested against several types of glioblastoma cells, demonstrating suitable results inhibiting cell viability, without normal cell side effects [22].

A great deal of attention has been focused in recent years on exploring the anticancer activity of biocompounds. Besides the anti-proliferative properties of these compounds, when used in patients, a diminished record of side effects are present, compared to chemotherapy or hormone treatment [23]. With this in mind, we assess the anticancer activity present in extracts from the brown alga *Egregia menziesii* in normal Bergmann glia cell line from chick, and cancer glioma cell lines from mouse, rat and human. We observed that *E. menziesii*’s extracts decrease cell viability of glioma cells without causing damage of normal glia cells.

## 2. Results

### 2.1. Cytotoxic Activity of E. menziesii’s Extracts over Artemia Salina Brine Shrimp

The first step to assess the cytotoxic potential of *E. menziesii* extracts was through the well-established *Artemia salina* brine shrimp toxicity test. Two way ANOVA showed that there was a significant effect of the *E. menziesii*’s extracts on the viability of *A. salina* (α = 0.05, F_(3, 24)_ = 380.1, *p* < 0.0001), concentrations (α = 0.05, F_(2, 24)_ = 1590, *p* < 0.0001), and the interaction between *E. menziesii*’s extracts and concentrations (α = 0.05, F_(6, 24)_ = 205.6, *p* < 0.0001) after 24 h of exposure. Specifically, Figure 1A shows differences in lethal activity profile among extracts. Hx and Chl extracts inhibited the viability of *A. salina* shrimp in a dose-dependent manner, whereas exposure of shrimps to MeOH extract caused death in 20–30% with all tested concentrations. The Hx and Chl extracts showed similar lethality pattern, the LC_50_ values for Hx and Chl extracts were in the range of the LC_50_ of l-ascorbic acid (LAA), a positive control, LC_50_ = 169.46 ± 0.067, 197.39 ± 0.015, and 150.33 ± 0.036 μg/mL, respectively. In contrast, the LC_50_ of MeOH extract was significantly higher than 1000 μg/mL (Figure 1B).

### 2.2. Determination of Cytotoxic Activity of E. menziesii’s Extracts against Nervous System Cell Lines

We evaluated the extracts in four brain cancer cell lines: *Mus musculus* neuroblastoma (N1E-115); *Rattus norvegicus* glioma (C6); human glioblastoma (U737) and immortalized human Müller cells (MIO-M1). A Bergmann glia cell primary culture was used as normal cell control. The cytotoxic activity over the nervous system cancer cell lines (C6, MIO-M1, N1-115, and U737) of Hx (α = 0.05, F_(4, 30)_ = 1.813 × 10^6^, *p* < 0.0001), Chl (α = 0.05, F_(4, 30)_ = 1.502 × 10^7^, *p* < 0.0001), and MeOH (α = 0.05, F_(4, 30)_ = 2.625 × 10^7^, *p* < 0.0001) extracts obtained from *E. menziesii* at different time of incubation (4 h: α = 0.05, F_(2, 30)_ = 4441, *p* < 0.0001; 24 h: α = 0.05, F_(2, 30)_ = 10,579, *p* < 0.0001; 48 h: α = 0.05, F_(2, 30)_ = 2.625 × 10^7^, *p* < 0.0001), and the interaction of cell lines and time of incubation (α = 0.05, F_(8, 30)_ = 1.493 × 10^6^, *p* < 0.0001; α = 0.05, F_(8, 30)_ = 7303, *p* < 0.0001; α = 0.05, F_(8, 30)_ = 2.625 × 10^7^, *p* < 0.0001) is summarized in Table 1. The estimation of the IC_50_ demonstrated that Hx and Chl extracts were the most effective. Moreover, the growth inhibitory effect of Hx and Chl extracts seems to be cell-specific, since its effect was observed mainly against C6, MIO-M1, and U737 cell lines. Particularly, the Hx extract proved to be significantly effective inhibiting the growth of the rat glioblastoma C6 cell line. Exposure of C6 cells to Hx extract, for 4, 24 and 48 h, significantly diminished their viability from 40 up to 90%, even above the inhibition observed in the group treated with 100 μM H_2_O_2_, that was 80% (data not shown). The IC_50_ of the Hx extract in C6 cells remains constant after 4 (9.51 ± 1.45 μg/mL), 24 (9.59 ± 1.34 μg/mL), or 48 h (8.59 ± 0.93 μg/mL) of exposure time (Table 1). On the contrary, the effectiveness of Hx extract against MIO-M1 cells increases as a function of the exposure time: after 4 (IC_50_ > 1000 μg/mL), 24 (IC_50_: 88.48 ± 1.65 μg/mL), and 48 h (IC_50_: 10.08 ± 1.98 μg/mL). The growth of N1-115 cells after 4 h of exposure to the Hx extract was IC_50_: 10.94 ± 1.93 μg/mL and surprisingly, no effects were observed in these cells at 24 and 48 h of exposure time (IC_50_ > 1000 μg/mL, for both extracts). In contrast, cytotoxicity was no observed neither in the U737 cell line nor in Bergmann glia cells (IC_50_ > 1000 μg/mL).

The Chl extract was effective by inhibiting the growth of C6, MIO-M1 and U737 cell lines. The highest effectiveness of Chl extract was observed in the C6 cell line, it inhibited cell growth with the same significant efficacy for all examined time periods: IC_50_: 9.82 ± 0.83 μg/mL, 8.86 ± 1.23 μg/mL, and 7.39 ± 1.43 μg/mL, after 4, 24 and 48 h of treatment, respectively. In MIO-M1 cells, the growth inhibition activity of Chl extract increased significantly with respect to the exposure time as observed in its IC_50_ after 4 (90.11 ± 1.23 μg/mL), 24 (86.32 ± 1.39 μg/mL), and 48 h (9.41 ± 1.93 μg/mL). The effect of Chl extract on U737 cells after being exposed during 4 (IC_50_: 105.71 ± 1.83 μg/mL), 24 (IC_50_: 108.85 ± 1.93 μg/mL), and 48 h (IC_50_: 95.76 ± 1.35 μg/mL) is also observed in Table 1. Chl extract did not show no significant effect in Bergmann glia cells (IC_50_ > 1000 μg/mL). Assays with the MeOH extract inhibited the growth of N1-115 cell line just after 48 h of treatment (IC_50_: 10.23 ± 1.23 μg/mL). MeOH extract did not show effectiveness inhibiting any other cell line. As observed in Table 1, the Hx and Chl extracts were the most effective inhibiting the growth of brain cancer cells. To our knowledge, there are no reports about the antineoplastic activity of seaweeds on brain tumor cells.

### 2.3. Cytotoxic Activity of E. menziesii Extracts over Human Peripheral Blood Lymphocytes (HPBL)

For many years, it has been accepted that cancer cells proliferate faster than normal cells; therefore most cancer drugs based therapies are design to target high-rate growing cells [24]. However, normal cells that grown fast (e.g., bone narrow, hair follicle, gastrointestinal and epidermal cells) are also largely affected by chemotherapy. Therefore, we evaluated the extracts using human peripheral blood lymphocytes (HPBL) through trypan-blue exclusion tests. In the assays, the same extract concentrations of *A. salina* test were used. Statistical analysis demonstrate that there was a significant effect of the *E. menziesii*’s extracts on the viability of HPBL (α = 0.05, F_(3, 24)_ = 491.1, *p* < 0.0001), concentrations (α = 0.05, F_(2, 24)_ = 714.3, *p* < 0.0001), and the interaction between *E. menziesii*’s extracts and concentrations (α = 0.05, F_(6, 24)_ = 137.0, *p* < 0.0001) after 1 h of exposure. Figure 2A shows that all extracts reduced HPBL viability in a dose-dependent manner. The Hx and Chl extracts inhibited up to 100% of HPBL growth, and the decreased in cell number resulted dose-dependent (Figure 2). In particular, a decrease of 20 and 30% was detected at 5 μg/mL, around 70% at 50 μg/mL, and 100% inhibition at 500 μg/mL, after Hx and Chl exposure. Again, these extracts showed similar effects (IC_50_ = 91.19 μg/mL and 71.5 μg/mL, respectively); whereas MeOH extract was unable to inhibit more than 30% at 500 μg/mL, and its LC_50_ was greater than 1000 μg/mL.

### 2.4. Cytotoxic Activity of E. menziesii Extracts against Differentiated and Non-Differentiated 3T3-L1 Cell Lines

Considering the cytotoxic activity results of *E. menziesii*’s extracts on HPBL could be an undesirable effect, we used 3T3-L1 fibroblastic cell line (derived from mouse embryo) to evaluate the toxicity of our extracts. This cell line can differentiate from fibroblast to adipose cells, moving from a fast growth state to a confluent and contact inhibited state [26], through the adipose differentiation with insulin and dexamethasone. Two way ANOVA showed that there was a significant effect of the *E. menziesii* extracts against differentiated (α = 0.05, F_(2, 18)_ = 41,369, *p* < 0.0001) and non-differentiated (α = 0.05, F_(2, 18)_ = 146,390, *p* < 0.0001) 3T3-L1 cell lines, time of incubation (α = 0.05, F_(2, 18)_ = 38,052, *p* < 0.0001; α = 0.05, F_(2, 18)_ = 585,555, *p* < 0.0001, respectively), and the interaction between *E. menziesii*’s extracts and time of incubation (α = 0.05, F_(4, 18)_ = 16,919, *p* < 0.0001; α = 0.05, F_(4, 18)_ = 146,390, *p* < 0.0001, respectively). As can be observed in Table 2, *E. menziesii* extracts were most effective inhibiting adipocytes than fibroblast. Again, the Hx and Chl extracts exhibit greater cell growth inhibitory activity of 3T3-L1 cells than the MeOH extract. However, the extracts inhibited mainly differentiated cells, which display slow growing profile. Additionally, MeOH extract did not inhibit the growth of non-differentiated 3T3-L1 cells, and it diminished the viability of differentiated cells after 48 h of treatment (IC_50_ = 107.79 ± 2.76 μg/mL). Therefore, we consider that the extracts contain a moderate anti-proliferative activity against 3T3-L1 fibroblast and adipose cells. However, it is important to perform a most extensive characterization of the anticancer properties of our extracts, using commercial cell panels [9].

## 3. Discussion

Cancer therapies represent an important challenge to clinicians, pharmaceutical companies and researchers, due to the large variety of cancer cell behavior within patients. Recently, seaweeds have attracted significant interest due to the diversity of species that display important anticancer activity [9]. We evaluated brown seaweed *E. menziesii*, since some species of its family have shown anticancer activity [16] using different cancer glioma cell lines from mouse, rat and, human as well as normal glia cells from chick.

The *A. salina* toxicity test represents a rapid (24 h), inexpensive, and simple assay to evaluate extracts and pure compounds, from natural and/or synthetic origin [27]; using a large number of organisms that allows statistical validation; in order to establish their potential cytotoxicity [28]. The results of this type of assay have been compared with other cytotoxicity tests, such as MTT assay to calculate lethal doses [29], and extrapolate the results to acute toxicity tests in mice [30]. Moreover, *A. salina* shrimps are resistant to several toxic agents (e.g., heavy metals) [31]. Therefore, this model constitutes an affordable and straightforward method to assess the toxicity of extracts and pure compounds. Accordingly, the results obtained in this study show that the Hx and Chl extracts inhibited up to 100% of *A. saline*’s growing, and therefore have to be considered as cytotoxic agents.

Brain cancers are difficult to treat due to the diversity of cells that comprise brain tumors, the high degree of malignancy and spreading potential [32,33]. Nowadays, there are not effective therapies against brain tumors. Radiation, chemotherapy and surgery, are the most common treatments for this type of malignancy [34]. Our findings are noteworthy since gliomas are the most common malignant primary tumors in CNS, which molecular characteristics and modifications make them difficult to treat [35]. For example, patients presenting glioblastoma multiform, that have been subject to surgery following of chemo-radiation and chemotherapy, have a median survival of only 14 months [36]. It is well known that glioma cells overexpress anti-oxidant enzymes that correlate with resistance to chemotherapeutic drugs [32], since the increase of anti-oxidant enzymes has been considered an adaptive mechanism of cancer cells against chronic stress [37]. This fact is relevant, in this study although we use glioma cancer cell lines from mouse, rat and human, our results showed that Hx and Chl extracts are effective by diminishing the viability of glioma cell lines (Table 1).

Usually antineoplastic treatments are unspecific, therefore both healthy and cancer cells are killed, and this sometimes leads to irreversible organ damage and the development of tolerance to the treatment [38,39]. Marine origin compounds are demonstrated to have promising selective anticancer activity [40,41]. Interestingly, our extracts did not diminish the viability of all cell lines tested. When the C6 cell line was treated, the IC_50_ of Hx and Chl extracts was around 10 μg/mL, at 4, 24 and 48 h of treatment. However, further human cell lines are needed to test the selectivity of the extracts. The carotenoid fucoxanthin diminished cell viability of human glioma cell lines U251 and U87, by promote apoptosis, inhibit migration and invasion activities on these cells. Moreover, fucoxanthin reduced the weight and volume of glioma mass in mice [42]. Likewise, the brown seaweed-isolated compound aplysin suppresses T98G cells invasion through Akt pathway inhibition [43]. However, future analysis is required to explore the metabolites associated with *E. menziessi*’s effects.

Interestingly, Bergmann glia viability was not altered with these same treatments. This inhibition activity profile was also observed in other seaweeds, e.g., *E. cava*, *E. bycils*, and *U. pinnatifida*, all of them members of the *Lessonaceae* family, which showed cell-specific activity against MCF7, HeLa and HepG2, respectively, without damaging normal cells, used as controls [40,44,45]. This effect was also observed over other glioma cells treated with seaweeds derivate compounds; that reduced cell viability of cancer cell lines without causing damage in normal cells [42,43,46]. Additionally, the methanol-dichlorometane (7:3) extract of *E. menziesii* did not show cytotoxic activity against HCT-11 colon cancer cells [47]. It has been described that several compounds derivate from seaweeds have important cytotoxic activity against brain cancer cell lines.

It is also noteworthy that the MeOH extract is non-toxic to any cell line tested (IC_50_ > 1000 μg/mL), except for N1-115 cells after 48 h of treatment, which IC_50_ was similar to those of Hx and Chl extracts (Table 1). Unexpectedly, the growth brain cell lines derived from human were not inhibited after the treatment with the extracts.

Even though we do not have evidence about the possible mechanism that mediates the cytotoxic activity of our extracts, our findings are not associated to a genotoxic event, since trypan-blue exclusion and MTT methods, evaluate the permeability of cell membrane and the metabolic rate of the cell, respectively. This is important to mention since several drugs used in chemotherapy are genotoxic agents, therefore it has been described that these drugs cause more severe DNA damage to cancer cells, mainly because of their weaker response to DNA damage and impaired DNA repair mechanisms, whereas in normal cells these drugs would lead to a severe growth arrest and cell death. Also, it has been suggested that this effect is less evident using general cytotoxic agents [48]. Even though it has been described that seaweeds can induce DNA fragmentation and apoptosis [49], there are other mechanisms through which seaweeds mediate their anticancer activity, including cell arrest, p53-dependent and independent apoptosis, increase of the antioxidant cell system, among others [9]. Specifically, the cytotoxic activity of seaweeds on glioma cells is due to suppression of invasion, inhibition of angiogenesis, migration; as well as induction of cell arrest, apoptosis and DNA fragmentation [42,43,46,50]. Furthermore, it has described that these effects are mediated mainly by the inhibition of the Akt pathway [43,50]. However, other signaling pathways such as the mitogen activated protein kinases pathway (MAPK), are also modulated by seaweeds [50,51]. Additionally, fucoidan induces the phosphorylation of p38 MAPK and inducible nitric oxide synthase expression in C6 glioma expression, which contributes with the anti-inflammatory response against neuronal damage [52].

Still, we can hypothesize about the possible compounds that could be responsible of the cytotoxic activity. It has been reported that several compounds such as fucoidans, laminarians, terpenoids, and polyphenols stated to possess anticancer, anti-tumor and anti-proliferative properties, are abundantly produced in brown seaweeds [27]. Particularly, the compounds that have shown cytotoxic activity against glioma cells are fucoidan [52,53]; the polyphenol eckol [50]; the carotenoid fucoxanthin [42]; aplysin [43]; phloroglucinol derivative 2,4-bis(4-fluorophenylacetyl) phloroglucinol [54]; and pheophorbide a [46]. Moreover, polyphenols have anti-oxidant effects that could be acting against glioma cell proliferation, similar to those effects observed in resveratrol treatments [32]. Furthermore, oxylipins were shown to have anti-cancer activity against several cancer cell types [55]. Taki-Nakano and colleagues demonstrated that the oxylipin 12-oxo phytodienoic acid has cytoprotective effect against human neuroblastoma SH-SY5Y cells, through Nrf2 signaling activation; that protects the cells from ROS-mediated cell death [56]. Oxylipins have been previously isolated from *E. menziesii* [57]. “However, future analysis is required in order to explore the metabolites associated with *E. menziesii* anti-proliferative effects”.

The delivery of anticancer drugs to brain tumors represents an important challenge for commercial drug development. The anticancer compounds isolated from seaweeds possessed complex chemical structures, which [57] in fact, might not cross the brain blood barrier (BBB). However, it has been proposed that several compounds from seaweeds (e.g., polysaccharides, fucoidans, and polyphenols with sugar substituents) are absorbed by the intestine through the SGLT1 and GLUT2 transport systems [58]. Therefore, we believe that this could be a possible mechanism to deliver the extracts through the BBB. Still, it is compulsory to probe this hypothesis using our extracts in an in vivo model.

Although a large percentage of commercial synthetic drugs have been developed after the discovery of the biological activities of natural products, it is also known that the development of new drugs, especially those intended for use in cancer therapies are expensive, usually take long time and frequently they are ineffective after the clinical trials. Thereby, prevention could be the main application of marine natural products like seaweeds [59,60].

In conclusion, the Hx and Chl extracts of *E. menziesii* seaweed possess growth inhibitory activity against rat C6 and human MIO-M1 cells. Therefore, this work open the possibility to study the mechanisms involved in the anticancer activity of seaweeds, against brain cancer cells, and the development of potential brain tumors drugs.

## 4. Materials and Methods

### 4.1. Drugs and Chemicals

Lymphoprep was purchased from Nycomed Pharma (Zürich, Switzerland) and tissue culture reagents from Gibco Invitrogen (Gaithersburg, MD, USA). All other chemicals were obtained from Sigma-Aldrich (St. Louis, MO, USA).

### 4.2. Animals

Chick embryos (10 days old) were kindly donated from Avi-Mex (Mexico City, Mexico) and maintained at 37 °C until used. All experiments were performed following the international guidelines on the ethical use of animals, under the specific approval of the Animal Ethics Committee of Cinvestav-Mexico, protocol number 0012-12. All efforts were made to reduce the number of embryos used and their suffering.

### 4.3. Seaweed Samples and Preparation of Extracts

Brown seaweed specimens of *Egregia menziesii* were collected in “*Campo No. 5*”, Punta Banda, Baja California, Mexico (31°44′3.45″ N; 116°43′39.95″ W) during August 2014. Algae were collected according to normative of Diario Oficial de la Federación [61], and in accordance with Instituto Nacional de Pesca (INAPESCA) Ensenada. Genus and specie of seaweeds were taxonomically verified by Oceanologist Luis Aguilar Rosas. All collected samples were gently rinsed with fresh water (to remove salt, sand and epiphytes), and then air-dried at 25 ± 3 °C in the course of 15 days. *E. menziesii*’s stipes, fronds, and pneumatocysts were separated manually and compiled. Extracts were prepared soaking the stipes and fronds in hexane, chloroform or methanol (1:9 *w*/*v*), for 20 days at room temperature. After, the solvent was removed under reduced pressure using a rotary vacuum evaporator (R300 model, Büchi, Flawil, Switzerland), light-green liquid fractions were obtained, and lyophilized (Lyobeta 15 model, Telstar, Terrassa, Spain); the obtained dark-green powder was stored at 4 °C. The yields of the three extracts are shown in Table 3.

### 4.4. Cytotoxic Activity by Brine Shrimp Lethality Test

A brine shrimp lethality bioassay was carried out to elucidate the cytotoxicity of seaweed extracts against *Artemia salina* nauplii, according to an established methodology [62] with minor modifications. Dried cysts were incubated (1 g cyst L^−1^) in artificial seawater at 27–30 °C for 24 h with strong aeration, under a continuous light regime. For toxicity test, hatched nauplii were collected with a pipette and concentrated in a small vial. Every single assay consisted of exposing groups of 10 nauplii to 5, 50 or 500 μg/mL of each of the three *E. menziesii*’s extracts. l-Ascorbic acid (LAA) and artificial seawater were used as positive and negative controls, respectively. Toxicity was determined after 24 h exposure by counting the number of living organisms, and calculating the percentage of mortality. Nauplii larvae were considered dead if they did not show any movement after 30 s of observing them under the stereoscope (C-Leds SMZ445 model, Nikon, Tokio, Japan), and their mortality percentage and the LC_50_ were calculated by linear regression analysis. Mortality below 50% was considered non-cytotoxic; mortality higher than 50% but below 75% was considered mildly cytotoxic; and mortality higher than 75% was considered highly cytotoxic according to the criteria of Vinayak et al. [27]. Assays were carried out in triplicate.

### 4.5. Lymphocyte Toxicity Test

Peripheral blood samples from healthy non-smokers donor volunteers (20–25 year-old) were collected by vein puncture according to standard procedures, and used within the 3 h from the collection. Informed consent was obtained in accordance with the Declaration of Helsinki. Human peripheral blood lymphocytes (HPBL) were isolated by centrifugation (centrifuge Beckman Coulter Allegra X-22R model, Brea, CA, USA) on Lymphoprep (Nycomed Pharma, Zürich, Switzerland) gradients, as described by the manufacturer. Extracts were dissolved in ethanol/water (3:1), and 5, 50 or 500 µg/mL of each extract were placed in triplicate in 24-well tissue culture plates. Thereafter, 1 × 10^5^ HPBL were seeded into these plates for 1 h at 37 °C. Partenolid (PrD, 10 mM in DMSO) and phosphate buffered saline (PBS) were used as positive and negative controls, respectively [25]. After incubation time, cells were stained with trypan blue and manually quantified in a Neubauer chamber. Cells’ mortality percentage and the LC_50_ were calculated by linear regression analyses.

### 4.6. Cell Culture

#### 4.6.1. Bergmann Glia Primary Cultures

Chick cerebellar Bergmann glia cultures were prepared as previously described [63]. Briefly, cerebella from 14-day-old chick embryos were dissected and homogenized mechanically. Cells were plated at a density of 8 × 10^5^ mL^−1^ in Dulbecco’s modified Eagle’s medium (DMEM) supplemented with 10% fetal bovine serum (FBS), 2 mM glutamine, 100 units/mL penicillin, and 100 µg/mL streptomycin. Cells were incubated (NU-5720 incubator, Nuaire, Plymouth, MN, USA) at 37 °C in 5% CO_2_ and used after 5–6 days in culture. As was demonstrated by Ortega et al. [63] immunocytochemical and kainate-induced ion fluxes of these primary cultures have shown that the vast majority of cultured cells are Bergmann glia cells. Confluent monolayers were treated as indicated above.

#### 4.6.2. Cell Lines

*Mus musculus* neuroblastoma (N1E-115), *Rattus norvegicus* glioma (C6), human glioblastoma (U737), and *Mus musculus* fibroblast (3T3-L1) cell lines were obtained from the American Tissue Culture Collection (ATCC, Manassas, VA, USA). The human Müller cell line Moorfields/Institute of Ophthalmology- Müller 1 (MIO-M1) was obtained from the UCL Institute of Ophthalmology (London, UK) [64]. Cells were grown and maintained at 37 °C with 5% of CO_2_ in DMEM supplemented with 10% FBS, 100 units/mL penicillin, and 50 µg/mL gentamicin. When the cells reached 80% of confluence, they were seeded at a density of 5 × 10^3^ mL^−1^ in sterile 96 well tissues culture plates for experimental treatment and analysis.

#### 4.6.3. In Vitro 3T3-L Cell Line Differentiation

A total of 5 × 10^3^ 3T3-L1 cells/well were seeded in complete DMEM medium (10% FCS), at a 80% of confluence the cells were supplemented with 0.25 μM dexamethasone and 10 μg/mL insulin for 3 days. Thereafter the cells and shifted to complete DMEM medium with 10 μg/mL insulin for 7 more days. Adipose differentiation was verified using Oil Red O staining according to Ramirez-Zacarias et al. [65].

### 4.7. In Vitro Viability Assays

Cell viability was determined by mitochondrial reduction of [3-(4,5-dimethylthiazol-2-yl)-2,5-diphenyltetrazolium bromide] (MTT) as described by Mosmann et al. [66]. Two days after reaching confluence, cells were treated during 4, 24 and 48 h with vehicle or 0.1, 1.0, 10, 100, 1000, and 1000 μg/mL of the seeded extracts. As positive control, the cells were exposed to 100 μM H_2_O_2_ for 15 min. At indicated time points, 50 μg/mL of MTT was added and incubated for 4 h. The media was removed and its absorbance was determined at 570 nm in a microplate reader (EPOCH, Biotek, Winooski, VT, USA). All plates were put back into the 37 °C incubator for 5 min, then transferred again to plate reader and their absorbance measured at 550 nm to determine the amount of total proteins per well. Data were represented as the mean ± SEM of OD at 570 nm, and were normalized with the amount of total protein. For each treatment, vehicle-stimulated cells were considered to have a 100% of viability; vehicles were each one of the same solvents used to do the extracts (hexane, chloroform and methanol), without seaweeds. Viability in vehicle-treated cells was not statistically different from non-treated cells (data not shown). Into the same plate assays were run by triplicate for each tested extract concentration. All experiments with cell lines were repeated at least 3 times.

### 4.8. Half Maximal Inhibitory Concentration (IC_50_) Determination

The IC_50_ was determined extrapolating the OD data obtained from the dose-response plot. The extract concentration that reduced the viability of cells by 50% (IC_50_) was determined by plotting triplicate data points over a concentration range and calculating values using the linear regression analysis function of the GraphPad PRISM Software version 5.00 for Mac (San Diego, CA, USA).

### 4.9. Statistical Analysis

Data are expressed as the mean ± SEM at least of three independent experiments. Homoscedasticity was evaluated in all data by D’Agostino-Pearson omnibus test. The data presented a normal distribution, and in all cases parametric analyzes were done. To determine differences between groups and the interaction effects of the extracts, we performed two-way ANOVA and Tukey’s test with multiple comparisons, α = 0.05, using PRISM Software version 5.00 for Mac (San Diego, CA, USA).

## Figures and Tables

**Figure 1 molecules-24-00260-f001:**
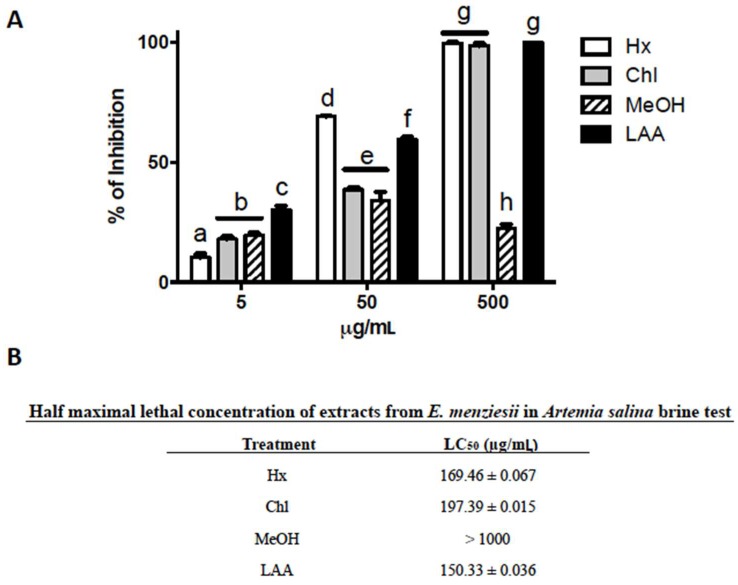
Cytotoxic activity of *E. menziesii*’s extracts over *Artemia salina* brine shrimp. (**A**) Percentage of brine shrimp death at 5, 50 or 500 μg/mL of hexane (Hx), chloroform (Chl) or methanol (MeOH) extracts obtained from *E. menziesii* seaweed for 24 h. l-Ascorbic acid (LAA) and artificial seawater were used as positive and negative controls, respectively. Data are expressed as the mean ± SEM of at least three independent experiments. Statistical differences were determined by two-way ANOVA, and differences among groups by Tukey’s test with multiple comparisons. a–g nomenclature: the same lowercase letters on each bar mean there are not differences among treatment values; conversely, distinct letters indicate significant differences within treatments (*p* < 0.01). (**B**) Half maximal lethal concentration (LC_50_) of Hx, Chl and MeOH extracts, and LAA was determined by linear regression.

**Figure 2 molecules-24-00260-f002:**
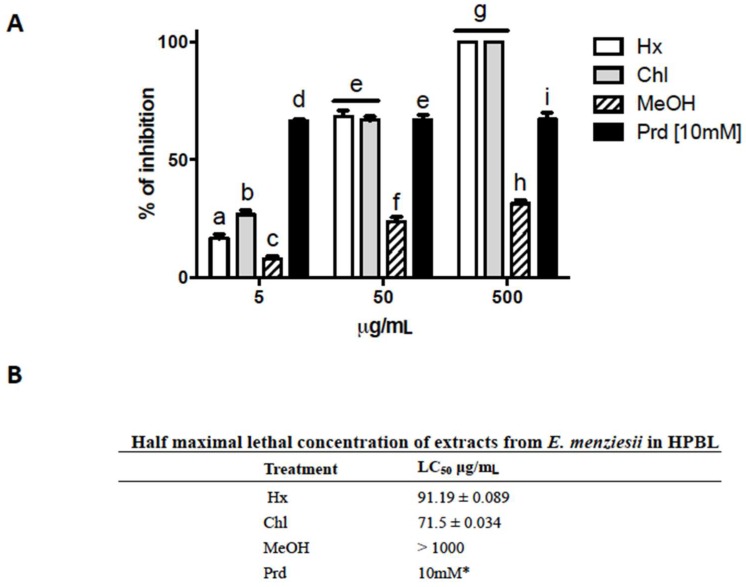
Cytotoxic activity of *E. menziesii*’s extracts over human peripheral blood lymphocytes (HPBL). (**A**) Percentage of growth inhibition of HPBL by the treatment with 5, 50 or 500 μg/mL of hexane (Hx), chloroform (Chl) or methanol (MeOH) extracts obtained from *E. menziesii* seaweed for 24 h. * 10 mM Partenolide [25], and PBS 1X were used as positive and negative controls, respectively. Data are expressed as the mean ± SEM at least of three independent experiments. Statistical differences were determined by two-way ANOVA, and differences between groups by Tukey’s test with multiple comparisons. a–i nomenclature: the same lowercase letters on each bar mean there are no significant differences among treatment values; conversely, distinct letters indicate significant differences within treatments. (**B**) Half lethal maximal concentration (LC_50_) of Hx, Chl and MeOH extracts, and LAA was determined by linear regression.

**Table 1 molecules-24-00260-t001:** Half maximal inhibitory concentration (IC_50_) [μg/mL] of the extracts of *E. menziesii* against nervous system cancer cell lines.

Cell Line	Hx			Chl			MeOH		
	4 h	24 h	48 h	4 h	24 h	48 h	4 h	24 h	48 h
BC	>1000 ^a^	>1000 ^c^	>1000 ^g^	>1000 ^a^	>1000 ^e^	>1000 ^i^	>1000 ^a^	>1000 ^b^	>1000 ^c^
C6	9.51 ± 1.45 ^b^	9.59 ± 1.34 ^d^	8.59 ± 0.93 ^h^	9.82 ± 0.83 ^b^	8.86 ± 1.23 ^f^	7.39 ± 1.43 ^j^	>1000 ^a^	>1000 ^b^	>1000 ^c^
MIO-M1	>1000 ^a^	88.48 ± 1.65 ^e^	10.08 ± 1.98 ^i^	90.11 ± 1.23 ^c^	86.32 ± 1.39 ^g^	9.41 ± 1.93 ^k^	>1000 ^a^	>1000 ^b^	>1000 ^c^
N1-115	10.94 ± 1.93 ^b^	>1000 ^c^	>1000 ^g^	>1000 ^a^	>1000 ^e^	>1000 ^i^	>1000 ^a^	>1000 ^b^	10.23 ± 1.23 ^d^
U737	>1000 ^a^	891.96 ± 1.23 ^f^	906.73 ± 1.73 ^j^	105.71 ± 1.83 ^d^	108.85 ± 1.93 ^h^	95.76 ± 1.35 ^l^	>1000 ^a^	>1000 ^b^	>1000 ^c^

BC: Bergmann glia primary culture cells; C6: *Rattus norvegicus* glioma cell line; MIO-M1: immortalized human Müller cell line; N1-115: *Mus musculus* neuroblastoma cell line; U737: human glioblastoma cell line; Hx: Hexanic extract; Chl: chloroformic extract; MeOH: methanolic extract. IC_50_ was determined as described in the Materials and Methods section. Two-way ANOVA and Tukey’s tests with multiple comparisons were performed for each extract tested, considering α value of 0.05. ^a–l^ nomenclature: the same superscript lowercase letters mean there are no significant differences among the values of each treatment; conversely, distinct letters indicate significant differences among treatments.

**Table 2 molecules-24-00260-t002:** Half maximal inhibitory concentration (IC_50_) [µg/mL] of the extracts of *E. menziesii* against 3T3-L1 cell line.

Extract	Fibroblast State			Adipose State		
	4 h	24 h	48 h	4 h	24 h	48 h
Hx	>1000 ^a^	>1000 ^a^	105.01 ± 2.12 ^b^	>1000 ^a^	110.15 ± 4.15 ^b^	8.78 ± 3.12 ^c^
Chl	>1000 ^c^	>1000 ^c^	107.79 ± 1.98 ^d^	13.17 ± 3.13 ^d^	109.15 ± 3.11 ^e^	8.46 ± 4.16 ^d^
MeOH	>1000 ^e^	>1000 ^e^	>1000 ^e^	>1000 ^f^	>1000 ^f^	107.79 ± 2.67 ^g^

Hx: Hexane extract; Chl: chloroform extract; MeOH: methanol extract. To determine IC_50_ see Materials and Methods section for details. Two-way ANOVA and Turkey’s test with multiple comparisons were performed for each cell state, considering α value of 0.05. ^a–g^ Same superscript lowercase letters mean there are no statistically significant differences between the values of each treatment; distinct letters indicate statistically significant differences between treatments.

**Table 3 molecules-24-00260-t003:** Yields of extracts obtained from *Egregia menziesii*.

Extract	Obtained Amount (g)	Yield (%)
Hx	1.48	0.25
Chl	2.96	0.49
MeOH	79.65	13.27

Hx: Hexane extract; Chl: chloroform extract; MeOH: methanol extract.

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
