# Peer review of "Brown Seaweed *Egregia menziesii*’s Cytotoxic Activity against Brain Cancer Cell Lines"

_molecules, 2019, doi:10.3390/molecules24020260_

Round 1
Reviewer 1 Report
The authors of the manuscript “Brown seaweed Egregia menziesii’s cytotoxic activity against brain cancer cell lines” used organic solvents (chloroform, methanol, and hexane) to prepare extracts from a brown seaweed and to examine in vitro cytotoxic effects of extracts on a variety of cell lines. The text is well-written and provides a clear message about possible anticancer effects of chloroform and hexane extracts. The methods are adequate and the results are presented properly. I would mention mostly technical issues which can be considered as a part of the minor revision of this manuscript:
1) Section “4.6. In vitro viability assays” needs specific clarification how the amount of proteins was measured. The current description is not clear. Which specific proteins assay was used?
2) Results of statistical analysis are shown for Tukey’s test on bar graphs as letters, which need to be explained in the figure legends. Also, two-way ANOVA test is mentioned, however no ANOVA stats are reported, i.e. F and P values for all data sets and interactions.
3) There are different types of technical issues such as missing Greek letters with concentration and typos (e.g. lines 157, 163, 242, 307 –Sigma=Sigma-Aldrich?, 323= italic, 473-473=use small letters, 569 = need full citation).
Author Response
Point 1: Section “4.6. In vitro viability assays” needs specific clarification how the amount of proteins was measured. The current description is not clear. Which specific proteins assay was used?
Response 1: We corrected the description and add the technique used to determine the amount of proteins. We did not describe the technique since Bradford is a universal and well-known protein assay method.
Point 2: Results of statistical analysis are shown for Tukey’s test on bar graphs as letters, which need to be explained in the figure legends. Also, two-way ANOVA test is mentioned, however no ANOVA stats are reported, i.e. F and P values for all data sets and interactions.
Response 2: According to the reviewer suggestion, the explanation of Tukey’s test letters was written in legends of those tables and bar graphs where these letters appeared. ANOVA stats are reported now, and both F and P values are included either in text or in figure legends.
Point 3: There are different types of technical issues such as missing Greek letters with concentration and typos (e.g. lines 157, 163, 242, 307 –Sigma=Sigma-Aldrich?, 323= italic, 473-473=use small letters, 569 = need full citation).
Response 3: The reviewer's observations were made and the missing Greek letters were corrected. In the text all mentioned genus and species were also corrected and written in italics or underlined, to differentiate them from the rest of the text. Letters size and full citation were changed too.

Reviewer 2 Report
This paper refers to the the anticancer activity of the brown algae Egregia menziesii’s extracts on glioma cells without causing damage of normal cells.
The work brings new knowledge to the antineoplastic activity of seaweeds on brain tumor cells. taking into account other articles that refer to the extraction, analysis and biological studies of brown seaweeds.
The cytotoxic potential of E. menziesii extracts was evaluated through the Artemia salina brine shrimp and against four brain cancer cell lines, peripheral blood lymphocytes and fibroblastic ell line.
The results seems to be interesting for readers but in order to improve the manuscript a small discussion about the possible chemical composition of the extracts corelated with the antitumor activity would be interesting.
In the section Materials and methods they do not show anything about type of different apparatus used in study, (microplate reader, flow cytometry, fluorescence microscope, nucleic acid analyzer, etc)if any.
Author Response
Point 1: The results seems to be interesting for readers but in order to improve the manuscript a small discussion about the possible chemical composition of the extracts corelated with the antitumor activity would be interesting.
Response 1: In the discussion, we stated: “fucoidans, laminarians, terpenoids, and polyphenols stated to possess anticancer, anti-tumor and anti-proliferative properties, are abundantly produced in brown seaweeds [26]. Particularly, the compounds that have shown cytotoxic activity against glioma cells are fucoidan [51, 52]; the polyphenol eckol [49]; the carotenoid fucoxanthin [41]; aplysin [42]; phloroglucinol derivative 2,4-bis (4-fluorophenylacetyl) phloroglucinol [53]; and pheophorbide a [45]. Also, we add the possible participation of oxylipins: “Furthermore, oxylipins are shown to have anti-cancer activity against several cancer cell types [55]. Taki-Nakano and colleagues demonstrated that the oxylipin 12-oxo phytodienoic acid has cytoprotective effect against human neuroblastoma SH-SY5Y cells, through Nrf2 signalling activation; that protects the cells from ROS-mediated cell death [56]. Oxylipins have been previously isolated from E. menziesii [57].”
Point 2: In the section Materials and methods they do not show anything about type of different apparatus used in study, (microplate reader, flow cytometry, fluorescence microscope, nucleic acid analyzer, etc)if any.
Response 2: The apparatus were mentioned within the methods section. Accordingly, in section 4.2. Seaweed samples and preparation of extracts: “rotary vacuum evaporator (BUCHI, R300 model), light-green liquid fractions were obtained, and lyophilized (Telstar, Lyobeta 15 model)”.
In section 4.3. Cytotoxic activity by brine shrimp lethality test: “movement after 30 sec of observing them under the stereoscope (Nikon C-Leds SMZ445)”.
In section 4.4. Lymphocyte toxicity test: “Human peripheral blood lymphocytes (HPBL) were isolated by centrifugation (centrifuge Beckman Coulter Allegra X-22R) on Lymphoprep (Nycomed Pharma) gradients, as described by the manufacturer.”
In section 4.5.1. Bergmann glia primary cultures: “Cells were incubated (incubator Nuaire NU-5720)”.
In section: 4.6. In vitro viability assays: “removed and its absorbance was determined at 570 nm in a microplate reader (EPOCH, Biotek).”

Reviewer 3 Report
In the manuscript titled “Brown seaweed E. menziesii’s……………………..cell lines” by Olivares-Banuelos et al., the authors showed that the extracts from the seaweed have cytotoxic and antiproliferative properties. They have used human and mouse brain cancer cell lines as well as the normal human cells like lymphocytes and fibroblasts to assess the effects this seaweed’s extracts. According to the present study, the extracts are good candidate for cancer prevention and novel chemotherapeutics. However, the conclusions have been extrapolated without supporting experimental evidences. The following are the comments: 1. As the authors mention that there are several studies on brown seaweed extracts having medicinal properties. So, what are the advantages of using E. menziesii over the other brown seaweed those have been studied so far? 2. Also, the authors mention that there are multiple metabolites from the brown seaweed including E. menziesii in the body. They may have different biological activities in the body. However, this study does not specify which metabolite or active component of the extract that may bring the anticancer effects. 3. In the viability tests, the treatments were compared and normalized to the untreated control as the 100 % vibility. It is essential to see the effect of the vehicle (Hexane, MeOH, Chloroform) which were used to derive the extract into. Therefore, the authors must show the vehicle control values and normalize it to the vehicle control than the untreated control. 4. The authors used only one human glioblastoma cell line to understand the effect of the extract. To conclude that the extract has an effect on the glioblastoma cells, more human cells must be investigated. Also, the effect on the human neurons must be presented to assess the effect on the normal brain cells. 5. When the authors assessed the effects of the extract in undifferentiated versus differentiated cells, the extract worked better on differentiated 3T3-L1 fibroblast cells. This contradicts the initial hypothesis that the cancer treatment should be specific to fast dividing cells and not damage body’s normal cells. Also, the experiment was done by using a mouse cell line. The experimental system used as well as the conclusions are far-fetched. 6. From line 199-208, the authors discuss the selectivity of the extract as it inhibits certain cells and not all the cells tested. However, the conclusions were based on a mouse cell line C6. We all agree that selectivity is an essential criterion but the authors tested only one human glioma cell line. Selectivity is important between tumor and normal cells and can also be beneficial if there is preferential effect on certain tumor subtypes than the others. Here in the context of this study, without assays performed on multiple human cell lines, the conclusions are extrapolated without evidence. 7. The last many paragraphs of discussion on the dietary component and nutritional values of brown seaweed E. menziesii is outside the scope of this manuscript. Also, without any experimental or population based statistical studies, it is beyond the context of this manuscript.
Author Response
Point 1: As the authors mention that there are several studies on brown seaweed extracts having medicinal properties. So, what are the advantages of using E. menziesii over the other brown seaweed those have been studied so far?
Response 1: The reviewer raises the pertinent question about the advantages of using E. menziesii over other brown seaweed. E. menziesii possesses an ecological advantage over the other brown seaweed since it is a North American located species, and most of the seaweeds described thus far to contain anti-cancer (and any other) activity are species present in Asia, specially India, China, Japan, although Brazilian algae have also been documented to present anti-cancer activity. There are very few studies about seaweeds from North America. It should be empathized the extensive metabolic differences between cold and hot water species of algae that impacts metabolites production. Moreover, E. menziesii is very abundant, and it reaches up to 15 meters long, and its fertility stage runs from spring to autumn (Henkel and Murray 2007; Kraemer and Chapman 1991). We certainly believe that these are interesting advantages that suggest commercial potential E. menziesii medicinal properties.
Point 2: Also, the authors mention that there are multiple metabolites from the brown seaweed including E. menziesii in the body. They may have different biological activities in the body. However, this study does not specify which metabolite or active component of the extract that may bring the anticancer effects.
Response 2: The referee points out that we do not specify which is the active component of the extract that possesses the anti-cancer activity. As we have outlined in the manuscript, seaweeds produce several metabolites that have been associated with their anti-cancer activity. While the reviewer is correct in his comment, in the discussion, we clearly mention that: “fucoidans, laminarians, terpenoids, and polyphenols display anticancer, anti-tumor and anti-proliferative properties, and are abundantly produced in brown seaweeds [26]. Particularly, the compounds that have shown cytotoxic activity against glioma cells are fucoidan [51, 52]; the polyphenol eckol [49]; the carotenoid fucoxanthin [41]; aplysin [42]; phloroglucinol derivative 2,4-bis (4-fluorophenylacetyl) phloroglucinol [53]; and pheophorbide a [45]. Also, we mention the plausible involvement of oxylipins: “Furthermore, oxylipins were shown to have anti-cancer activity against several cancer cell types [55]. Taki-Nakano and colleagues demonstrated that the oxylipin 12-oxo phytodienoic acid has cytoprotective effect against human neuroblastoma SH-SY5Y cells, through Nrf2 signalling activation; that protects the cells from ROS-mediated cell death [56]. Oxylipins have been previously isolated from E. menziesii [57].” Oxylipins are derived from polyunsaturated fatty acids, therefore their hydrocarbon chains are very large, and so they could be extracted with hexane, which is slightly polar, although also chloroform could extract them.
Accordingly, in line 294 we add: “However, future analysis is required in order to explore the metabolites associated with E. menziessi anti-proliferative effects”.
Point 3: In the viability tests, the treatments were compared and normalized to the untreated control as the 100 % viability. It is essential to see the effect of the vehicle (Hexane, MeOH, Chloroform) which were used to derive the extract into. Therefore, the authors must show the vehicle control values and normalize it to the vehicle control than the untreated control.
Response 3:
The reviewer is worried about a possible effect of the vehicle used in the extract in terms of cell viability. As described in the Methodology, the vehicle-treated cells are set to a 100% viability. Nevertheless, in order to clarify the point in the revised manuscript we have specified that viability in vehicle-treated cells was not statistically different from non-treated cells (data not shown).
Point 4: The authors used only one human glioblastoma cell line to understand the effect of the extract. To conclude that the extract has an effect on the glioblastoma cells, more human cells must be investigated. Also, the effect on the human neurons must be presented to assess the effect on the normal brain cells.
Response 4: The reviewer is worried about the fact that we only assay the anti-proliferative activity of our extracts in one cell line and therefore our conclusions might be not being correct. He also prompts us to expose human neuros to assess the effect of the extracts in normal brain cells. It is not clear for us this latter suggestion; how do we get a human neuronal primary culture? If the reviewer is thinking of the various human neuronal cell lines commercially available, we tend to differ from this option, since these cells have already a disruption in their proliferative properties that precisely enable them to divide. As control cells, we have used a well-characterized Bergmann glia primary culture, and we have done so keeping in mind that glia cells outnumber neurons by a factor of ten in most brain areas like the cerebellum, which by the way is an excellent model of normal cells since it contains more than half of the total brain neurons in vertebrates. Besides, it has been established that gliomas are the most aggressive brain tumors. In terms of using more than one glioma cell line and check if our extracts are capable to inhibit their proliferation, we agree with the referee, although being this an explorative study, we will certainly continue to work with our extracts and try to isolate the compound(s) bearing the anti-cancer properties and these compounds will certainly be assayed in more transformed cell lines. In any event, we have softened our conclusions.
Point 5: When the authors assessed the effects of the extract in undifferentiated versus differentiated cells, the extract worked better on differentiated 3T3-L1 fibroblast cells. This contradicts the initial hypothesis that the cancer treatment should be specific to fast dividing cells and not damage body’s normal cells. Also, the experiment was done by using a mouse cell line. The experimental system used as well as the conclusions are far-fetched.
Response 5: The reviewer is surprised that our extracts are more effective in differentiated fibroblasts, compromising our initial hypothesis. Although we could agree with this reviewer’s comment, it should be noted that differentiated fibroblasts have lost cell division capacity, clearly due to a substantial difference in their protein repertoire. Is the reason of their differential susceptibility? We tend to favor this interpretation; obviously, at this stage we do not know which are the molecular targets of the anti-proliferative compounds present in our extracts. It is quite possible that our extracts have lipogenic effects (we also observed the diminished amount of total lipids) that favor differentiation. It is clear that more studies, outside the scope of this contribution are needed to clarify this point.
Point 6: From line 199-208, the authors discuss the selectivity of the extract as it inhibits certain cells and not all the cells tested. However, the conclusions were based on a mouse cell line C6. We all agree that selectivity is an essential criterion but the authors tested only one human glioma cell line. Selectivity is important between tumor and normal cells and can also be beneficial if there is preferential effect on certain tumor subtypes than the others. Here in the context of this study, without assays performed on multiple human cell lines, the conclusions are extrapolated without evidence.
Response 6: Again, the reviewer raises the point of the use of multiple human cell lines. As already discussed in response 4, we have softened our conclusions. As the reviewer, might be aware, disruption in cell proliferation control is a multifactorial molecular event and is different not only in terms of the different cell types within an organism, but also completely different form individual to individual. This is the molecular frame of the impossibility to efficiently control certain tumor progressions in human beings. Is the complexity of human cancer a barrier to try to identify compounds that could potentially inhibit cell proliferation? We do not think so. The effectiveness of a certain compound will certainly be different from individual to individual so at this stage, it is not surprising that our extracts have different capacity to inhibit cell growth. Regarding the fact that we use non-human derived cell lines, this has been done for years and the reason is again, the heterogeneity of human tumors. Nevertheless, to settle this argument, in the revised manuscript we now claim: “Interestingly, our extracts did not diminish the viability of all cell lines tested. When the C6 cell line was treated, the IC50 of Hx and Chl extracts was around 10 μg/ml, at 4, 24 and 48h of treatment. However, further human cell lines are needed to test the selectivity of the extracts.”
Point 7: The last many paragraphs of discussion on the dietary component and nutritional values of brown seaweed E. menziesii is outside the scope of this manuscript. Also, without any experimental or population based statistical studies, it is beyond the context of this manuscript.
Response 7: We agree with the reviewer’s comment. Thereby, we eliminated this part of the discussion, since do not fit with the scope of the journal.

Round 2
Reviewer 3 Report
In the revised version of the manuscript and the answers provided by the authors have addressed the issues raised. The manuscript can be accepted in its current version.